# CAN AN IMAGE CLASSIFIER SUFFICE FOR ACTION RECOGNITION?

**Quanfu Fan,**[*] **Chun-Fu (Richard) Chen,**[*] **Rameswar Panda**[*]
MIT-IBM Watson AI Lab
qfan@us.ibm.com, chenrich@us.ibm.com, rpanda@ibm.com

## ABSTRACT

We explore a new perspective on video understanding by casting the video recognition problem as an image recognition task. Our approach rearranges input video frames into super images, which allow for training an image classifier directly to fulfill the task of action recognition, in exactly the same way as image classification. With such a simple idea, we show that transformer-based image classifiers alone can suffice for action recognition. In particular, our approach demonstrates strong and promising performance against SOTA methods on several public datasets including Kinetics400, Moments In Time, Something-Something V2 (SSV2), Jester and Diving48. We also experiment with the prevalent ResNet image classifiers in computer vision to further validate our idea. The results on both Kinetics400 and SSV2 are comparable to some of the best-performed CNN approaches based on spatio-temporal modeling. Our source codes and models are available at https://github.com/IBM/sifar-pytorch.

## 1 INTRODUCTION

The recent advances in convolutional neural networks (CNNs) (He et al., 2016; Tan & Le, 2019), along with the availability of large-scale video benchmark datasets (Kay et al., 2017; Monfort et al., 2019; Damen et al., 2020), have significantly improved action recognition, one of the fundamental problems of video understanding. Many existing approaches for action recognition naturally extend or borrow ideas from image recognition. At the core of these approaches is *spatio-temporal modeling*, which regards time as an additional dimension and jointly models it with space by extending image models (i.e., 3D CNNs) (Tran et al., 2015; Carreira et al., 2017; Feichtenhofer, 2020) or fuses temporal information with spatial information processed separately by 2D CNN models (Lin et al., 2019; Fan et al., 2019). CNN-based approaches demonstrate strong capabilities in learning saptio-temporal feature representations from video data.

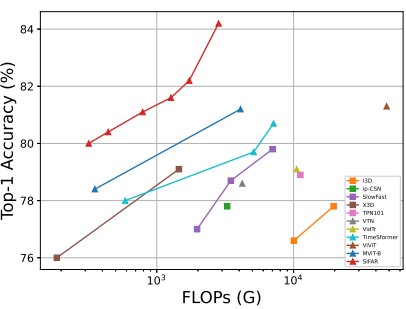

Figure 1: Comparison of our proposed SI-FAR (red) with SOTA approaches for action recognition on Kinetics400 dataset.

Videos present long-range pixel interactions in both space and time. It's known in approaches like non-local networks (Wang et al., 2018) that modeling such relationships helps action recognition. The recently emerging Vision Transformers (ViTs) naturally own the strength of capturing long-range dependencies in data, making them very suitable for video understanding. Several approaches (Bertasius et al., 2021a; Li et al., 2021; Arnab et al., 2021) have applied ViTs for action recognition and shown better performance than their CNN counterparts. However, these approaches are still following the conventional paradigm of video action recognition, and perform temporal modeling in a similar way to the CNN-based approaches using dedicated self-attention modules.

In this work, we explore a different perspective for action recognition by casting the problem as an image recognition task. We ask if it is possible to model temporal information with ViT directly

---

[*]Equal contribution.

without using dedicated temporal modules. In other words, can an image classifier alone suffice for action recognition? To this end, we first propose a simple idea to turn a 3D video into a 2D image. Given a sequence of input video frames, we rearrange them into a super image according to a pre-defined spatial layout, as illustrated in Fig. 2. The super image encodes 3D spatio-temporal patterns in a video into 2D spatial image patterns. We then train an image classifier to fulfill the task of action recognition, in exactly the same way as image classification. Without surprise, based on the concept of super images, any image classifier can be re-purposed for action recognition. For convenience, we dub our approach *SIFAR*, short for **S**uper **I**mage **f**or **A**ction **R**ecognition.

We validate our proposed idea by using Swin Transformer (Liu et al., 2021), a recently developed vision transformer that has demonstrated good performance on both image classification and object detection. Since a super image has a larger size than an input frame, we modify the Swin Transformer to allow for full self-attention in the last layer of the model, which further strengthens the model's ability in capturing long-range temporal relations across frames in the super image. With such a change, we show that SIFAR produces strong performance against the existing SOTA approaches (Fig. 1) on several benchmark datsets including Kinetics400 (Kay et al., 2017), Moments in Time (Monfort et al., 2019), Something-Something V2 (SSV2) Goyal et al. (2017), Jester (Materzynska et al., 2019) and Diving48 (Li et al., 2018). Our proposed SIFAR also enjoys efficiency in computation as well as in parameters. We further study the potential of CNN-based classifiers directly used for action recognition under the proposed SIFAR framework. Surprisingly, they achieve very competitive results on Kinetics400 dataset against existing CNN-based approaches that rely on much more sophisticated spatio-temporal modeling. Since $3 \times 3$ convolutions focus on local pixels only, CNN-based SIFAR handles temporal actions on Something-Something less effectively. We experiment with larger kernel sizes to expand the temporal receptive field of CNNs, which substantially improves the CNN-based SIFAR by $4\% - 6.8\%$ with ResNet50.

SIFAR brings several advantages compared to the traditional spatio-temporal action modeling. Firstly, it is simple but effective. With one single line of code change in PyTorch, SIFAR can use any image classifier for action recognition. We expect that similar ideas can also work well with other video tasks such as video object segmentation (Duke et al., 2021). Secondly, SIFAR makes action modeling easier and more computationally efficient as it doesn't require dedicated modules for temporal modeling. Nevertheless, we do not tend to underestimate the significance of temporal modeling for action recognition. Quite opposite, SIFAR highly relies on the ability of its backbone network to model long-range temporal dependencies in super images for more efficacy. Lastly, but not the least, the perspective of treating action recognition the same as image recognition unleashes many possibilities of reusing existing techniques in a more mature image field to improve video understanding from various aspects. For example, better model architectures (Tan & Le, 2019), model pruning (Liu et al., 2017) and interpretability (Desai & Ramaswamy, 2020), to name a few.

## 2 RELATED WORK

**Action Recognition from a Single Image.** One direction for video action recognition is purely based on a single image (Davis & Bobick, 1997; Zhao et al., 2017; Safaei & Foroosh, 2019; Bilen et al., 2016). In (Davis & Bobick, 1997), multiple small objects are first identified in a still image and then the target action is inferred from the relationship among the objects. Other approaches such as (Safaei & Foroosh, 2019) propose to predict the missing temporal information in still images and then combine it with spatial information for action classification. There are also approaches that attempt to summarize RGB or motion information in a video into a representative image for action recognition, e.g., motion-energy image (MEI) (Davis & Bobick, 1997), Dynamic Image Network (Bilen et al., 2016), Informative Frame Synthesis (IFS) (Qiu et al., 2021), Adaptive Weighted Spatio-temporal Distillation (AWSD) (Tavakolian et al., 2019b) and Adversarial Video Distillation (AVD) (Tavakolian et al., 2019a). Nonetheless, our approach does not attempt to understand a video from a single input image or a summarized image. Instead our proposed approach composites the video into a super image, and then classifies the image with an image classifier directly.

**Action Recognition with CNNs.** Action recognition is dominated by CNN-based models recently (Feichtenhofer et al., 2018; Carreira et al., 2017; Fan et al., 2019; Feichtenhofer, 2020; Chen et al., 2021; Lin et al., 2019; Wang et al., 2016; Zhou et al., 2018; Liu et al., 2020; Jiang et al., 2019a; Tran et al., 2019). These models process the video as a cube to extract spatial-temporal

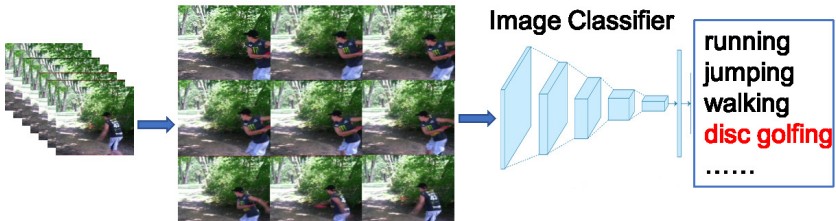

Figure 2: **Overview of SIFAR**. A sequence of input video frames are first rearranged into a super image based on a $3 \times 3$ spatial layout, which is then fed into an image classifier for recognition.

features via the proposed temporal modeling methods. E.g., SlowFast (Feichtenhofer et al., 2018) proposes two pathways with different speed to capture short-range and long-range time dependencies. TSM (Lin et al., 2019) applies a temporal shifting module to exchange information between neighboring frames and TAM (Fan et al., 2019) further enhances TSM by determining the amount of information to be shifted and blended. On the other hand, another thread of work dynamically attempts to select the key frame of an activity for faster recognition (Wu et al., 2019; 2020; Meng et al., 2020; 2021; Sun et al., 2021). E.g., Adaframe (Wu et al., 2019) employs a policy network to determine whether or not this is a key frame, and the main network only processes the key frames. ARNet (Meng et al., 2020) determines what the image resolution should be used to save computations based on the importance of input frame images. Nonetheless, our approach is fundamentally different from conventional action recognition. It simply uses an image classifier as a video classifier by laying out a video to a super image without explicitly modeling temporal information.

**Action Recognition with Transformer.** Following the vision transformer (ViT) (Dosovitskiy et al., 2021), which demonstrates competitive performance against CNN models on image classification, many recent works attempt to extend the vision transformer for action recognition (Neimark et al., 2021; Li et al., 2021; Bertasius et al., 2021b; Arnab et al., 2021; Fan et al., 2021). VTN (Neimark et al., 2021), VidTr (Li et al., 2021), TimeSformer (Bertasius et al., 2021b) and ViViT (Arnab et al., 2021) share the same concept that inserts a temporal modeling module into the existing ViT to enhance the features from the temporal direction. E.g., VTN (Neimark et al., 2021) processes each frame independently and then uses a longformer to aggregate the features across frames. On the other hand, divided-space-time modeling in TimeSformer (Bertasius et al., 2021a) inserts a temporal attention module into each transformer encoder for more fine-grained temporal interaction. MViT (Fan et al., 2021) develops a compact architecture based on the pyramid structure for action recognition. It further proposes a pooling-based attention to mix the tokens before computing the attention map so that the model can focus more on neighboring information. Nonetheless, our method is straightforward and applies the Swin (Liu et al., 2021) model to classify super images composed from input frames.

Note that the joint-space-time attention in TimeSformer (Bertasius et al., 2021a) is a special case of our approach since their method can be considered as flattening all tokens into one plane and then performing self-attention over all tokens. However, the memory complexity of such an approach is prohibitively high, and it is only applicable to the vanilla ViT (Dosovitskiy et al., 2021) without inductive bias. On the other hand, our SIFAR is general and applicable to any image classifiers.

## 3 APPROACH

### 3.1 OVERVIEW OF OUR APPROACH

The key insight of SIFAR is to turn spatio-temporal patterns in video data into purely 2D spatial patterns in images. Like their 3D counterparts, these 2D patterns may not be visible and recognizable by human. However, we expect they are characteristic of actions and thus identifiable by powerful neural network models. To that end, we make a sequence of input frame images from a video into a super image, as illustrated in Fig. 2, and then apply an image classifier to predict the label of the video. Note that the action patterns embedded in a super image can be complex and may involve both local (i.e., *spatial information* in a video frame) and global contexts (i.e., *temporal dependencies* across frames). It is thus understandable that effective learning can only be ensured by image classifiers with

strong capabilities in modeling short-range and long-range spatial dependencies in super images. For this reason, we explore the recently developed vision transformers based on self-attention to validate our proposed idea. These methods come naturally with the ability to model global image contexts and have demonstrated competitive performance against the best-performed CNN-based approaches on image classification as well as action recognition. Next we briefly describe Swin Transformer (Liu et al., 2021), an efficient approach that we choose to implement our main idea in this work.

**Preliminary.** The Vision Transformer (ViT) [13] is a purely attention-based classifier borrowed from NLP. It consists of stacked transformer encoders, each of which is featured with a multi-head self-attention module (MSA) and a feed-forward network (FFN). While demonstrating promising results on image classification, ViT uses an isotropic structure and has a quadruple complexity w.r.t image resolution in terms of memory and computation. This significantly limits the application of ViT to many vision applications that requires high-resolution features such as object detection and segmentation. In light of this issue, several approaches (Liu et al., 2021; Chu et al., 2021; Zhang et al., 2021) have been proposed to perform region-level local self-attention to reduce memory usage and computation, and Swin Transformer is one of such improved vision transformers.

**Swin Transformer** (Liu et al., 2021) first adopts a pyramid structure widely used in CNNs to reduce computation and memory. At the earlier layers, the network keeps high image resolution with fewer feature channels to learn fine-grained information. As the network goes deeper, it gradually reduces spatial resolution while expanding feature channels to model coarse-grained information. To further save memory, Swin Transformer limits self-attention to non-overlapping local windows (W-MSA) only. The communications between W-MSA blocks is achieved through shifting them in the succeeding transformer encoder. The shifted W-MSA is named as SW-MSA. Mathematically, the two consecutive blocks can be expressed as:

$$\begin{aligned}
\mathbf{y}_k &= \text{W-MSA}(\text{LN}(\mathbf{x}_{k-1})) + \mathbf{x}_{k-1}, \\
\mathbf{x}_k &= \text{FFN}(\text{LN}(\mathbf{y}_k)) + \mathbf{y}_k, \\
\mathbf{y}_{k+1} &= \text{SW-MSA}(\text{LN}(\mathbf{x}_k)) + \mathbf{x}_k, \\
\mathbf{x}_{k+1} &= \text{FFN}(\text{LN}(\mathbf{y}_{k+1})) + \mathbf{y}_{k+1},
\end{aligned} \tag{1}$$

where $\mathbf{x}_l$ is the features at the $l^{th}$ layer and FFN and LN are feed-forward network and layer normalization, respectively.

**SIFAR.** In our case, SIFAR learns action patterns by sliding window, as illustrated in Fig. 3. When the sliding window (blue box) is within a frame, spatial dependencies are learned. On the other hand, when the window (red box) spans across frames, temporal dependencies between them are effectively captured. The spatial pooling further ensures longer-range dependencies across frames captured.

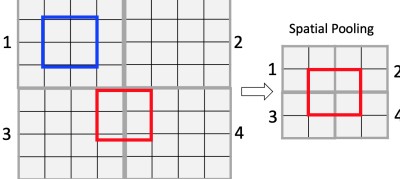

Figure 3: Swin Transformer does self-attention in a local window. In SIFAR, when the window (blue box) is within a frame, spatial dependencies are learned within a super image (4 frames here). When the window spans across different frames (red box), temporal dependencies between them are effectively captured. The spatial pooling further ensures longer-range dependencies to be learnt. Best viewed in color.

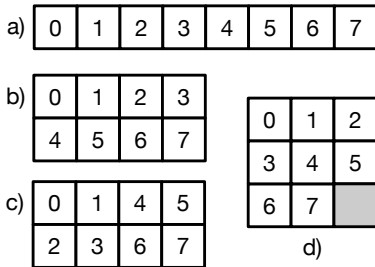

Figure 4: **Grid Layout**. We apply a grid to lay out the input frames. Illustrated here are several possible layouts for 8 frames, i.e., a) $1 \times 8$, b) and c) $2 \times 4$, and d) $3 \times 3$, respectively. Empty images are padded at the end if grid is not full.

**Creation of Super Image.** Given a set of video frames, we order them by a given layout (Fig. 4) to form a large super image. Different layouts give different spatial patterns for an action class. We hypothesize that a more compact structure such as a square grid may facilitate a model to learn temporal dependencies across frames as such a shape provides the shortest maximum distance between any two images. Given $n$ input frames, we create a super image by placing all the frames in

order onto a grid of size $(m-1) \times m$ when $n < (m-1) \times m$ or $m \times m$ when $n \geq (m-1) \times m$ where $m = \lceil \sqrt{n} \rceil$. Empty images are padded at the end if the grid is not full. With this method, for example, 12 frames will be fit into a $3 \times 4$ grid while 14 frames into a $4 \times 4$ grid. In the default setting, we use a $3 \times 3$ layout for 8 images and a $4 \times 4$ one for 16 images, respectively. There are other spatial arrangements as well (see Fig. 4 for more examples). However our experiments empirically show that a square grid performs the best across different datasets. Note that our approach has linear computational complexity w.r.t the number of input frames. As described above, the size of a super image is $m$ ($m = \lceil \sqrt{n} \rceil$) times as large as the size of a frame image, suggesting that the total number of tokens (or image patches) in Swin grows linearly by $n$.

**Sliding Window.** As previously mentioned, Swin Transformer performs self-attention within a small local window to save memory. It uses a uniform window size across all layers, and the default window size is 7 in the original paper. Since a larger window leads to more interactions across frames, which is beneficial for SIFAR to learn long-range temporal dependencies in super images, we slightly modify the architecture of Swin Transformer (Liu et al., 2021) for it to take different window sizes flexibly in self-attention. In particular, we keep the same window size for all the layers except the last one, whose window is as large as its image resolution, implying a global self-attention including all the tokens. Since the last layer has only two transformer encoders, the computational overhead imposed by an increased window size is quite small, as indicated in Table 1.

Table 1: **Model architectures of SIFAR**. The number in a model name indicates the window size used by the model before the last layer. "B" means Swin-B. † denotes the models using 16 frames as input and ‡ indicates the models using a larger input image resolution.

| Model | Frames | Image Size | FLOPs (G) | Window Size |
|---|---|---|---|---|
| SIFAR-B-7 | 8 | 224 | 138 | {7,7,7,7} |
| SIFAR-B-12 | 8 | 192 | 106 | {12,12,12,18} |
| SIFAR-B-14 | 8 | 224 | 147 | {14,14,14,21} |
| SIFAR-B-12† | 16 | 192 | 189 | {12,12,12,24} |
| SIFAR-B-14† | 16 | 224 | 263 | {14,14,14,28} |
| SIFAR-B-12‡ | 8 | 384 | 423 | {12,12,12,36} |

The change of window size may result in adjustment of the input image size as the image resolution at each layer must be divisible by the window size in Swin Transformer. As noted in Table 1, SIFAR-B-7 keeps the vanilla architecture of Swin-B. SIFAR-B-12 is more efficient than SIFAR-B-7 because SIFAR-B-12 takes smaller images ($192^2$) as input. We demonstrate later in Sec. 4 that a larger window is critical for SIFAR to achieve good performance on more temporal datasets such as SSV2.

**Implementation.** Once the spatial layout for the input frames is determined, implementing our idea in PyTorch is as simple as inserting into an image classifier the following line of code, which changes the input of a video to a super image.

```
# create a super image with a layout (sh, sw) pre-specified by the user.
x = rearrange(x, 'b c (sh sw) h w -> b c (sh h) (sw w)', sh=sh, sw=sw)
```

The trivial code change described above transforms an image classifier into an video action classifier. Our experiments show that the same training and evaluation protocols for action models can be still applied to the repurposed image classifier.

## 4 EXPERIMENTS

### 4.1 DATASETS AND EXPERIMENTAL SETUP

**Datasets.** We use Kinetics400 (K400) (Kay et al., 2017), Something-Something V2 (SSV2) (Goyal et al., 2017), Moments-in-time (MiT) (Monfort et al., 2019), Jester (Materzynska et al., 2019), and Diving48 (Li et al., 2018) datasets in our evaluation. Kinetics400 is a widely-used benchmark for action recognition, which includes ∼240k training videos and 20k validation videos in 400 classes. SSV2 contains 220k videos of 174 types of predefined human-object interactions with everyday objects. This dataset is known for its high temporal dynamics. MiT is a fairly large collection of one million 3-second labeled video clips, involving actions not only from humans, but also from animals, objects and natural phenomena. The dataset includes around 800k training videos and 33,900 validation videos in 339 classes. Jester contains actions of predefined hand gestures, with 118,562 and 14,787 training and validation videos over 27 classes, respectively. Diving48 is an action recognition dataset without representation bias, which includes 15,943 training videos and 2,096 validation videos over 48 action classes.

**Training**. We employ *uniform sampling* to generate video input for our models. Such a sampling strategy divides a video into multiple segments of equal length, and has shown to be effective on both Kinetics400 and SSV2 Chen et al. (2021). We train all our models by finetuning a Swin-B model (Liu et al., 2021) pretrained on ImageNet-21K (Deng et al., 2009), except for those SSV2 models, which are fine tuned from the corresponding Kinetics400 models in Table 3.

Our training recipes and augmentations closely follow DeiT (Touvron et al., 2020). First, we apply multi-scale jitter to augment the input (Wang et al., 2016) with different scales and then randomly crop a target input size (e.g. 8×224×224 for SIFAR-B-7). We then use Mixup (Zhang et al., 2018) and CutMix (Yun et al., 2019) to augment the data further, with their values set to 0.8 and 1.0, respectively. After that, we rearrange the image crops as a super image. Furthermore, we apply drop path (Tan & Le, 2019) with a rate of 0.1, and enable label smoothing (Szegedy et al., 2016) at a rate of 0.1. All our models were trained using V100 GPUs with 16G or 32G memory. Depending on the size of a model, we use a batch size of 96, 144 or 192 to train the model for 15 epochs on MiT or 30 epochs on other datasets, including 5 warming-up epochs. The optimizer used in our training is AdamW (Loshchilov & Hutter, 2019) with a weight decay of 0.05, and the scheduler is Cosine (Loshchilov & Hutter, 2017) with a base linear learning rate of 0.0001.

**Inference.** We first scale the shorter side of an image to the model input size and then take three crops (top-left, center and bottom-right) for evaluation. The average of the three predictions is used as the final prediction. We report results by top-1 and top-5 classification accuracy (%) on validation data, the total computational cost in FLOPs and the model size in number of parameters.

## 4.2 MAIN RESULTS

**Comparison with Baselines.** We first compare our approach with several representative CNN-based methods including I3D (Carreira et al., 2017), TSM (Lin et al., 2019) and TAM (Fan et al., 2019). Also included in the comparison are two TimeSformer models (Bertasius et al., 2021a) based on the same backbone Swin-B (Liu et al., 2021) as used by our models. All the models considered take 8 frames as input. As can be seen from Table 2, our approach substantially outperforms the CNN baselines on Kinetics400 while achieving comparable results on SSV2. Our approach is also better than TimeSformer on both datasets. These results clearly demon-

Table 2: **Comparison with Baseline Methods.** All models use 8 frames as input.

| Model | SSV2 | | Kinetics400 | |
|---|---|---|---|---|
| | Top-1 | Top-5 | Top-1 | Top-5 |
| I3D-R50 | 61.1 | 86.5 | 72.6 | 90.6 |
| TSM-R50 | 59.1 | 85.6 | 74.1 | 91.2 |
| TAM-R50 | 62.0 | 87.3 | 72.2 | 90.4 |
| TimeSformer* | 35.9 | 71.1 | 77.5 | 92.5 |
| TimeSformer** | 58.7 | 85.9 | 80.1 | 94.4 |
| SIFAR-B-7 | 59.0 | 86.0 | 79.6 | 94.4 |
| SIFAR-B-12 | 60.8 | 87.3 | 80.0 | 94.5 |
| SIFAR-B-14 | 61.6 | 87.9 | 80.2 | 94.4 |

*: Swin-B (space only); **: Swin-B (divided space-time).

strate that a powerful image classifier like Swin Transformer can learn expressive spatio-temporal patterns effectively from super images for action recognition. In other words, an image classifier can suffice video understanding without explicit temporal modeling. The results also confirm that a larger sliding window is more helpful in capturing temporal dependencies on temporal datasets like SSV2. Our approach performs global self-attention in the last layer of a model only (see Table 1). This substantially mitigates the memory issue in training SIFAR models.

**Kinetics400.** Table 3 shows the results on Kinetics400. Our 8-frame models (SIFAR-12 and SIFAR-14) achieve 80.0% and 80.2% top-1 accuracies, outperforming all the CNN-based approaches while being more efficient than the majority of them. SIFAR-B-14† further gains ∼ 1.8% improvement, benefiting from more input frames. Especially, SIFAR-L-12‡ yields an accuracy of 84.2%, the best among all the very recently developed approaches based on vision transformers including TimeSformer (Bertasius et al., 2021b) and MViT-B (Fan et al., 2021). Our proposed approach also offers clear advantages in terms of FLOPs and model parameters compared to other approaches except MViT-B. For example, SIFAR-B-12‡ has 5× and 37× fewer FLOPs than TimeSformer-L and ViViT-L, respectively, while being 1.4× and 3.6× smaller in model size.

**SSV2.** Table 4 lists the results of our models and the SOTA approaches on SSV2. With the same number of input frames, our approach is 1 ∼ 2% worse than the best-performed CNN methods. However, our approach performs on par with other transformer-based method such as TimeSformer (Bertasius et al., 2021a) and VidTr-L (Li et al., 2021) under the similar setting. Note that ViViT-L (Arnab et al., 2021) achieves better results with a larger and stronger backbone ViT-L (Dosovitskiy et al., 2021). MViT-B (Fan et al., 2021) is an efficient multi-scale architecture,

Table 3: **Comparison with Other Approaches on Kinetics400.**

| Model | #Frames | Pretrain | Params(M) | FLOPs(G) | Top-1 | Top-5 |
|---|---|---|---|---|---|---|
| TSN-R50 (Wang et al., 2016) | 32 | IN-1K | 24.3 | 170.8×30 | 69.8 | 89.1 |
| TAM-R50 (Fan et al., 2019) | 32 | IN-1K | 24.4 | 171.5×30 | 76.2 | 92.6 |
| I3D-R50 (Carreira et al., 2017) | 32 | IN-1K | 47.0 | 335.3×30 | 76.6 | 92.7 |
| I3D-R50+NL (Wang et al., 2018) | 32 | IN-1K | − | 282×30 | 76.5 | 92.6 |
| I3D-R101+NL (Wang et al., 2018) | 32 | IN-1K | − | 359×30 | 77.7 | 93.3 |
| ip-CSN-152 (Tran et al., 2019) | 32 | − | 32.8 | 109×30 | 77.8 | 92.8 |
| SlowFast8×8 (Feichtenhofer et al., 2018) | 32 | − | 27.8 | 65.7×30 | 77.0 | 92.6 |
| SlowFast8×8+NL (Feichtenhofer et al., 2018) | 32 | − | 59.9 | 116×30 | 78.7 | 93.5 |
| SlowFast16×8+NL (Feichtenhofer et al., 2018) | 64 | − | 59.9 | 234×30 | 79.8 | 93.9 |
| X3D-M (Feichtenhofer, 2020) | 16 | − | 3.8 | 6.2×30 | 76.0 | 92.3 |
| X3D-XL (Feichtenhofer, 2020) | 16 | − | 11.0 | 48.4×30 | 79.1 | 93.9 |
| TPN101 (Yang et al., 2020) | 32 | − | | 374×30 | 78.9 | 93.9 |
| VTN-VIT-B (Neimark et al., 2021) | 250 | IN-21K | 114.0 | 4218×1 | 78.6 | 93.7 |
| VidTr-L (Li et al., 2021) | 32 | IN-21K | − | 351×30 | 79.1 | 93.9 |
| TimeSformer (Bertasius et al., 2021b) | 8 | IN-21K | 121.4 | 196×3 | 78.0 | - |
| TimeSformer-HR (Bertasius et al., 2021b) | 16 | IN-21K | 121.4 | 1703×3 | 79.7 | − |
| TimeSformer-L (Bertasius et al., 2021b) | 96 | IN-21K | 121.4 | 2380×3 | 80.7 | − |
| ViViT-L (Arnab et al., 2021) | 32 | IN-21K | 310.8 | 3992×12 | 81.3 | 94.7 |
| MViT-B (Fan et al., 2021) | 16 | − | 36.6 | 70.5×5 | 78.4 | 93.5 |
| MViT-B (Fan et al., 2021) | 64 | − | 36.6 | 455×9 | 81.2 | 95.1 |
| SIFAR-B-12 | 8 | IN-21K | 87 | 106×3 | 80.0 | 94.5 |
| SIFAR-B-12† | 16 | IN-21K | 87 | 189×3 | 80.4 | 94.4 |
| SIFAR-B-14 | 8 | IN-21K | 87 | 147×3 | 80.2 | 94.4 |
| SIFAR-B-14† | 16 | IN-21K | 87 | 263×3 | 81.8 | 95.2 |
| SIFAR-L-14† | 16 | IN-21K | 196 | 576×3 | 82.2 | 95.1 |
| SIFAR-B-12‡ | 8 | IN-21K | 87 | 423×3 | 81.6 | 95.2 |
| SIFAR-L-12‡ | 8 | IN-21K | 196 | 944×3 | **84.2** | **96.0** |

Table 4: **Comparison with Other Approaches on SSV2.**

| Model | #Frames | Params(M) | FLOPs(G) | Top-1 | Top-5 |
|---|---|---|---|---|---|
| TAM-R50 (Fan et al., 2019) | 8 | 24.4 | 42.9×6 | 62.8 | 87.4 |
| TAM-R50 (Fan et al., 2019) | 32 | 24.4 | 171.5×6 | 63.8 | 88.3 |
| I3D-R50 (Carreira et al., 2017) | 8 | 47.0 | 83.8×6 | 61.1 | 86.5 |
| I3D-R50 (Carreira et al., 2017) | 32 | 47.0 | 335.3×6 | 62.8 | 88.0 |
| TSM-R50 (Lin et al., 2019) | 8 | 24.3 | 32×6 | 59.1 | 85.6 |
| TSM-R50 (Lin et al., 2019) | 16 | 24.3 | 65×6 | 63.4 | 88.5 |
| TPN-R50 (Yang et al., 2020) | 8 | − | − | 62.0 | − |
| TAM-bLR101 (Fan et al., 2019) | 64 | 40.2 | 96.4×1 | 65.2 | 90.3 |
| MSNet (Kwon et al., 2020) | 16 | 24.6 | 67×1 | 64.7 | 89.4 |
| STM (Jiang et al., 2019b) | 16 | 24.0 | 67×30 | 64.2 | 89.8 |
| TEA (Liu et al., 2020) | 16 | − | 70×30 | 65.1 | 89.9 |
| TimeSformer (Bertasius et al., 2021b) | 8 | 121.4 | 196×3 | 59.5 | − |
| TimeSformer-HR (Bertasius et al., 2021b) | 16 | 121.4 | 1703×3 | 62.5 | − |
| ViViT-L (Arnab et al., 2021) | 32 | 100.7 | − | 65.4 | 89.8 |
| VidTr-L (Li et al., 2021) | 32 | − | − | 60.2 | − |
| MViT-B (Fan et al., 2021) | 16 | 36.6 | 70.5×3 | 64.7 | 89.2 |
| MViT-B (Fan et al., 2021) | 64 | 36.6 | 455×3 | **67.7** | **90.9** |
| SIFAR-B-12 | 8 | 87 | 106×3 | 60.8 | 87.3 |
| SIFAR-B-12† | 16 | 87 | 189×3 | 61.4 | 87.4 |
| SIFAR-B-14 | 8 | 87 | 147×3 | 61.6 | 87.9 |
| SIFAR-B-14† | 16 | 87 | 263×3 | 62.6 | 88.5 |
| SIFAR-L-14† | 16 | 196 | 576×3 | 64.2 | 88.4 |

which can process much longer input sequences to capture fine-grained motion patterns in SSV2 data. Training SIFAR models with more than 16 frames still remains computationally challenging, especially for models like SIFAR-B-14 and SIFAR-L-14†, which need a larger sliding window size. Our results suggest that developing more efficient architectures of vision transformer be an area of improvement and future work for SIFAR to take advantage of more input frames on SSV2.

**MiT.** MiT is a large diverse dataset containing label noise. As seen from Table 5, with the same backbone ViT-L, SIFAR-L-12‡ is ∼4% better than ViViT-L (Arnab et al., 2021), and outperforms AssembelNet (Ryoo et al., 2020) based on neural architecture search by a considerable margin of 8%.

**Jester and Diving48.** We further evaluate our proposed approach on two other popular benchmarks: Jester (Materzynska et al., 2019) and Diving48 (Li et al., 2018). Here we only consider the best single models from other approaches for fair comparison. As shown in Table 6, SIFAR achieves competitive

Table 5: **Comparison with Other Methods on MiT.**

| Model | Top-1 | Top-5 |
|---|---|---|
| TRN-Incpetion (Zhou et al., 2018) | 28.3 | 53.9 |
| TAM-R50 (Fan et al., 2019) | 30.8 | 58.2 |
| I3D-R50 (Chen et al., 2021) | 31.2 | 58.9 |
| SlowFast-R50-8×8 (Feichtenhofer et al., 2018) | 31.2 | 58.7 |
| CoST-R101 (Li et al., 2019) | 32.4 | 60.0 |
| SRTG-R3D-101 (Stergiou & Poppe, 2020) | 33.6 | 58.5 |
| AssembleNet (Ryoo et al., 2019) | 33.9 | 60.9 |
| ViViT-L (Arnab et al., 2021) | 38.0 | 64.9 |
| SIFAR-B-12‡ | 39.9 | 69.2 |
| SIFAR-L-12‡ | **41.9** | **70.3** |

Table 6: **Comparison with Other Approaches on Jester and Diving48.**

(a) Jester

| Model | Top-1 | Top-5 |
|---|---|---|
| TSN-Inception (Wang et al., 2016) | 95.0 | 99.9 |
| TRN-Inception (Zhou et al., 2018) | 95.3 | — |
| TSM-R50 (Lin et al., 2019) | 95.0 | 99.9 |
| PAN-R50 (Zhang et al., 2020) | 99.6 | 99.8 |
| STM-R50(Jiang et al., 2019a) | 96.7 | 99.9 |
| I3D-R50 (Carreira et al., 2017) | 96.4 | — |
| TAM-R50 (Fan et al., 2019) | 96.4 | — |
| SlowFast-R50-8×8 (Feichtenhofer et al., 2018) | 96.8 | — |
| SIFAR-B-12† | 97.2 | 99.9 |
| SIFAR-B-14† | **97.2** | **99.9** |

(b) Diving48

| Model | Top-1 | Top-5 |
|---|---|---|
| TimeSformer (Bertasius et al., 2021b) | 74.9 | — |
| TimeSformer-HR (Bertasius et al., 2021a) | 78.0 | — |
| TimeSformer-L (Bertasius et al., 2021a) | 81.0 | — |
| SlowFast (Feichtenhofer et al., 2018) | 77.6 | — |
| SIFAR-B-12† | 85.3 | 98.3 |
| SIFAR-B-14† | **87.3** | **98.8** |

results again on both datasets, surpassing all other models in comparison. Note that the Diving48 benchmark contains videos with similar background and objects but different action categories, and is generally considered as an unbiased benchmark. Our model SIFAR-B-14† outperforms TimeSformer-L by a large margin of 6% on this challenging Diving48 dataset.

**Classification by CNNs.** We also test our proposed approach using the ResNet image classifiers on both SSV2 and Kinetics400 datasets. For fairness, the ResNet models are pretrained on ImageNet-21K. Table 7 shows the results. Our models clearly outperform the traditional CNN-based models for action recognition on Kinetics400. Especially, with a strong backbone R152x2 (a model 2× wider than Resnet152), SIFAR-R152x2 achieves a superior accuracy of 79.0%, which is surprisingly comparable to the best CNN results (SlowFast16×8+NL: 79.8%) reported in Table 3.

Table 7: CNN-based SIFAR Results

| Model | # Frames | SSV2 | Kinetics400 |
|---|---|---|---|
| I3D-R50 (Carreira et al., 2017) | 8 | 61.1 | 72.6 |
| TSM-R50 (Wang et al., 2016) | 8 | 59.1 | 74.1 |
| TAM-R50 (Fan et al., 2019) | 8 | 62.0 | 72.2 |
| SIFAR-R50 | 8 | 50.8 | 73.2 |
| SIFAR-R101 | 8 | 56.3 | 76.6 |
| SIFAR-R152×2* | 8 | 58.2 | 79.0 |
| SIFAR-R50-C7 | 8 | 54.4 (+3.6) | 74.4 (+1.2) |
| SIFAR-R50-C11 | 8 | 55.2 (+4.2) | 74.5 (+1.3) |
| SIFAR-R50-C21 | 8 | 55.8 (+5.0) | 74.8 (+1.6) |
| SIFAR-R50-C21-11 | 8 | 57.6 (+6.8) | 75.1 (+1.9) |
| SIFAR-R101-C21 | 8 | 58.1 (+1.8) | 77.7 (+1.1) |
| SIFAR-R101-C21-11 | 8 | 59.6 (+3.3) | 77.5 (+0.9) |

*: a model two times wider than R152

On SSV2, the results of CNN-based SIFAR are less satisfactory but reasonable. This is because 3x3 convolutions are local with a small receptive field, thus failing to capturing long-range temporal dependencies in super images. We hypothesize that a larger kernel size with a wider receptive field may address this limitation and potentially improve the performance of CNN-based SIFAR models. To validate this, we perform additional experiments by adding one or two more residual blocks to the end of ResNet models with larger kernel sizes, i.e. replacing the second convolution in those new blocks by a 7x7, 11x11 or 21x21 kernel. These models are indicated by names ending with "C7" (7x7), "C11" (11x11) or "C21" (21x21) in Table 7. As seen from the table, using larger kernel sizes consistently improves the results on both ResNet50 and ResNet101 models. For example, we obtain an absolute 5.0% improvement over original ResNet50 and 2.0% over original ResNet101 respectively, using one more block with a kernel size of 21x21. When adding another block with a kernel size of 11x11 (i.e. SIFA-R50-C21-11 and SIFA-R101-C21-11), it further boosts the performance up to 6.8% with ResNet50 and 2.7% with ResNet101. These results strongly suggest that expanding the receptive field of CNNs be a promising direction to design better CNN-based SIFAR models.

Table 8: **Ablation Study.** The effects of each component on model accuracy.

(a) Super Image Layout. (SIFAR-B-12 on SSV2)

| Layout | Top-1 | Top-5 |
|---|---|---|
| 1×8 (Fig. 4a) | 44.4 | 74.4 |
| 2×4 (Fig. 4b) | 58.6 | 85.5 |
| 2×4 (Fig. 4c) | 58.1 | 85.1 |
| 3×3 (Fig. 4d) | 60.8 | 87.3 |

(b) Absolute Positioning Embedding.

| Model | SSV2 | | Kinetics400 | |
|---|---|---|---|---|
| | w/ APE | w/o APE | w/ APE | w/o APE |
| SIFAR-B-7 | 56.6 | 56.4 | 79.7 | 79.6 |
| SIFAR-B-12 | 60.8 | 59.5 | 79.7 | 80.0 |
| SIFAR-B-14 | 61.6 | 60.1 | 80.0 | 80.2 |

### 4.3 ABLATION STUDIES

**How does an image layout affect the performance?** The layout of a super image determines how spatio-temporal patterns are embedded in it. To analyze this, we trained a SIFAR model on SSV2 for each layout illustrated in Fig. 4. As shown in Table 8a, a strip layout performs the worst while a grid layout produces the best results, which confirms our hypothesis.

**Does absolute positioning embedding help?** Swin paper (Liu et al., 2021) shows that when relative position bias are added, Absolute Position Embedding (APE) is only moderately beneficial for classification, but not for object detection and segmentation. They thus conclude that inductive bias that encourages certain translation invariance is still important for vision tasks. To find out whether or not APE is effective in our proposed approach, we add APE to each frame rather than each token. The results in Table 8b indicate that APE slightly improves model accuracy on SSV2, but is harmful to Kinetics400. In our main results, we thus apply APE to SSV2 only.

**Does the temporal order of input matter?** We evaluate the SIFAR-B-12 model (trained with input frames of normal order) by using three types of input with different temporal orders, i.e the reverse, random and normal. As shown in Table 9, SIFAR is not sensitive to the input order on Kinetics400, whereas on SSV2, changing the input order results in a significant performance drop. This is consistent with the finding in the S3D paper (Xie et al., 2018), indicating that for datasets like SSV2 where there are visually similar action categories, the order of input frames matters in model learning.

Table 9: **Effects of temporal order.**

| Order | Kinetics400 | SSV2 |
|---|---|---|
| normal | 80.0 | 60.8 |
| reverse | 79.8 | 23.9 |
| random | 79.7 | 39.4 |

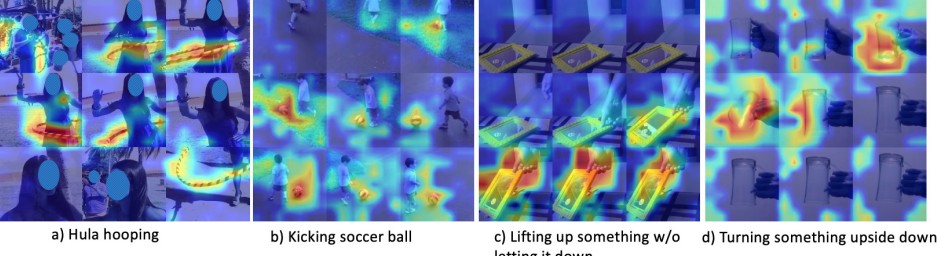

a) Hula hooping    b) Kicking soccer ball    c) Lifting up something w/o letting it down    d) Turning something upside down

Figure 5: **Visualization by Ablation CAM** (Desai & Ramaswamy, 2020)

**What does SIFAR learn?** We apply ablation CAM (Desai & Ramaswamy, 2020), an image model interpretability technique, to understand what our models learn. Fig. 5 shows the Class Activation Maps (CAM) of 4 actions correctly predicted by SIFAR-B-12. Not surprisingly, the model learns to attend to objects relevant to the target action such as the hula hoop in a) and soccer ball in b). In c) and d), the model seems to correctly focus on where meaningful motion happens.

## 5 CONCLUSION

In this paper, we have presented a new perspective for action recognition by casting the problem as an image recognition task. Our idea is simple but effective, and with one line of code to transform an sequence of input frames into a super image, it can re-purpose any image classifier for action recognition. We have implemented our idea with both CNN-based and transformer-based image classifiers, both of which show promising and competitive results on several popular publicly available video benchmarks. Our extensive experiments and results show that applying super images for video understanding is an interesting direction worth further exploration.

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
