# OpenReview forum: "Can an Image Classifier Suffice For Action Recognition?"
_ICLR.cc/2022/Conference — ICLR 2022 Poster_

### Official Review · Reviewer_K68r · 2021-10-17

**Correctness:** 3
**Technical Novelty And Significance:** 2
**Empirical Novelty And Significance:** 4
**Recommendation:** 8
**Confidence:** 3

**Main Review:**

### Strengths
- The proposed method (Super Image for Action Recognition, SIFAR) is extremely simple and easy to implement (one liner in pytorch). Its accuracy is on par with SOTA, and its speed is also SOTA given a fixed accuracy threshold.
- Taking deep networks' ability to model spatial relationships and then using it to model temporal relationships is a very interesting direction. This is an orthogonal approach than having explicit components for modeling temporal relationships. I was surprised that it worked so well.
- Being able to connect image classification and action recognition can enable image classification techniques to be applied to action recognition, which could potentially accelerate the field.
- The ablation studies on layout and ordering are interesting.
- The writing is very clear. The writing also clearly mentions the limitations, such as "*Training SIFAR models with more than 16 frames still remains computationally challenging, especially for models like SIFAR-B-14 and SIFAR-L-14†, which need a larger sliding window size.*"

### Weakness
- From the analysis done on SSV2, it seems like the bigger limitations of SIFAR are (1) it has difficulty taking in more than 16 images at once, and (2) it is unclear what is the limit of SIFAR in terms of doing fine-grained temporal modeling. However, I don't think these limitations prevent SIFAR from being an interesting idea.
- "*Note that the joint-space-time attention in TimeSformer (Bertasius et al., 2021a) is a special case of our approach since their method can be considered as flattening all tokens into one plane and then performing self-attention over all tokens. However, the memory complexity of such an approach is prohibitively high...*" I am not sure I agree with this claim. In figure 3 of (Bertasius et al., 2021a, https://arxiv.org/pdf/2102.05095.pdf), the method can use up to 96 frames with the "Divided Space-time" approach, which is less memory intensive than SIFAR, which has difficulty going more than 16 frames at once.
- In Table 9, SSV2 in reverse order leads to the worst performance, which is surprising. I would expect that normal and reverse ordering should lead to the same performance, as whether the first image is at the top left or bottom right should not affect the network’s ability to learn as long as if the network is given consistently ordered input. Can the authors elaborate on this? Thank you!


### Typos
- P5: “o”ur approach has linear computational complexity… (capitalize)
- P5: We demonstrate later in Sec. 4 “thata” larger window

**Summary Of The Paper:**

The paper proposes to perform action recognition by first rearranging the frames from a video into a 3x3 or 4x4 grid to form a "super image", and then giving the super image to a standard image classifier to perform action recognition. Given that this super image will be a larger image, the paper leverages the more memory efficient Swin Transformer [1] as an image classifier to perform action recognition. Experiments on Kinetics400, Moments In Time, Something-Something V2 (SSV2), Jester and Diving48 show that the proposed method is on par or exceeds SOTA in terms of accuracy. On Kinetics400, the method not only is SOTA in terms of accuracy, but also is the most FLOPs-efficient method given a specific accuracy. The strong performance suggests that a deep network's ability to model spatial relationships could also be applied to model temporal relationships across frames in a video, which is an orthogonal direction to having explicit components in the network modeling temporal relationships. Furthermore, being able to connect action recognition with image classification enables existing image classification techniques to be applied to action recognition, which could potentially accelerate the field.

[1]: Ze Liu, Yutong Lin, Yue Cao, Han Hu, Yixuan Wei, Zheng Zhang, Stephen Lin, and Baining Guo. Swin Transformer: Hierarchical Vision Transformer using Shifted Windows. arXiv.org, March 2021.

**Summary Of The Review:**

Overall, I think the proposed method, though simple, leads to surprisingly good results. It not only provides a new way of thinking about modeling temporal relationships, but also better connects action recognition and image classification. Therefore, even though there may not be as much technical novelty in the paper, I still vote for acceptance of the paper.

---

> ### Author Response · Authors · 2021-11-23
> **Response to Reviewer K68r**
>
> We thank the reviewer for his/her constructive comments and insightful suggestions. Below we provide a point by point reply.
>
> **(a) Fine-grained action classification** The Diving48 benchmark contains videos with similar background and objects but different action categories. It is generally considered as an unbiased benchmark as well as a fine-grained action dataset. In the future, we plan to
> evaluate our approach further on larger fine-grained action datasets such as EPIC-KITCHENS (https://epic-kitchens.github.io/2021) and FineGym (https://sdolivia.github.io/FineGym/).
>
> **(b) Memory complexity of joint-space-time attention in TimeSformer.** The 96-frame TimeSformer model is based on the divided space-time scheme, which separates spatial and temporal attention for action modeling. This scheme has a computational complexity of $O((N+K)^2)$, and it is significantly more efficient than the joint space-time scheme we mentioned in our paper, which has a complexity of $O((N*K)^2)$.  Here N is number of tokens per frame and K is the total number of input frames.
>
> **\(c\) Normal and reversal frame order.** In the experiment of Table 9, the SIFAR models were trained using input frames with normal temporal order, but tested by reversing and shuffling the input frames. In this case, SSV2 is shown to be sensitive to the input order. However, as correctly pointed out by the reviewer, when the input order is consistent in both training and test, the models tend to produce similar results regardless of what the  input order is. The table below confirms this.
>
>
> | Training order | Test order  | Kinetics400 | SSV2|
> | --- | --- | --- | --- |
> |normal | normal |80.0  | 60.1 |
> |random*| random* |79.7 | 60.4  |
> |normal | reverse | 79.8 | 23.9|
> |normal | random | 79.7 | 39.4 |
>
> \* the same order, but different from normal.
>
> **(d) Typos.** Thanks for pointing them out. We have fixed them in our paper.

---

> > ### Comment · Reviewer_K68r · 2021-11-26
> > **Thank you for the author's response.**
> >
> > Thank you for the author's response. Points (a), (c), and (d) are clear.
> >
> > For point (b), I am not sure if the original question was answered. Sorry that I did not ask it clearly initially. My question is: is the paragraph below fair when TimeSformer can process 96 frames at once while SIFAR can only do 9 frames? Thanks!
> >
> > "Note that the joint-space-time attention in TimeSformer (Bertasius et al., 2021a) is a special case of
> > our approach since their method can be considered as flattening all tokens into one plane and then
> > performing self-attention over all tokens. However, the memory complexity of such an approach
> > is prohibitively high, and it is only applicable to the vanilla ViT (Dosovitskiy et al., 2021) without
> > inductive bias. On the other hand, our SIFAR is general and applicable to any image classifiers"

---

> > > ### Author Response · Authors · 2021-11-29
> > > **clarification on point (b)**
> > >
> > > We thank the reviewer for the feedback!  Below we provide some further clarification on point (b).
> > >
> > > There are three attention schemes proposed in the TimeSformer paper, i.e. space-only, divided space-time and joint space-time. According to the paper (Section 4.1 and Figure 3),
> > > the **divided-space-time** scheme can scale up to 96 input frames while the **joint-space-time attention** scheme has GPU memory issues
> > > with 32 or longer input frames. We consider the **joint-space-time attention** scheme a special case of our approach as this scheme
> > > performs self-attention on the tokens from all the input frames, in a similar way to our approach. We agree that such a statement
> > >  might cause confusion given that TimeSformer and our approach are based on different backbones and have different complexity in memory and computation. We will clarify this in the final version of our paper. Please let us know if this helps or if you have any other questions or feedback.

---

> > > > ### Comment · Reviewer_K68r · 2021-11-30
> > > > **clarification on point (b)**
> > > >
> > > > Thank you for the reply.
> > > > Yes I think if we can clearly specify which scheme we are comparing against (joint-space-time attention, and not divided-space-time) in the text, then I think it would help avoid confusion. Thanks!
> > > >
> > > > The authors have addressed my concerns, and I vote for acceptance of the paper.

---

### Official Review · Reviewer_utfq · 2021-11-02

**Correctness:** 3
**Technical Novelty And Significance:** 2
**Empirical Novelty And Significance:** 2
**Recommendation:** 6
**Confidence:** 5

**Details Of Ethics Concerns:**

No concern.

**Main Review:**

Strength:
1) Transforming a video to an image for video recognition is a good direction to explore and has potential application field in the future;
2) This paper is well written and the experiment is extensive;
3) The visualization provides interesting insight of this task;

Weakness:
1) The novelty of the proposed method is limitted. The main contribution of this paper is frame re-arrangement strategy, which provides little inspiration to the community;
2) The size of the introduced super image is larger than the video frame. There are already some works focusing on transforming a video to an image, like [1]. Their generated images bear the same resolution as the video frame；
3) The involvement of the Swin-Transformer seems unreasonable. The local operation can only extract the boundary information of the consecutive frames in the super image. For example the information from the right boundary of frame 1 and the left boundary of frame 2, which can not be stated as the temporal dependency. In my understanding, modeling the temporal dependency is to caputure the variation of the similar region (for example the same object or human) across frames.The only part in the modified Swin-Transformer that is able to model temporal dependncy is the kernel with the same size of the image in the last layer;
4) The reference is not complete. The works like AWSD [2], AVD [3] are missing, which also considering to treat a video clip as an image;

Reference:
1) Qiu, Zhaofan, et al. "Condensing a Sequence to One Informative Frame for Video Recognition." Proceedings of the IEEE/CVF International Conference on Computer Vision. 2021.
2) Tavakolian, Mohammad, Hamed R. Tavakoli, and Abdenour Hadid. "Awsd: Adaptive weighted spatiotemporal distillation for video representation." Proceedings of the IEEE/CVF International Conference on Computer Vision. 2019.
3) Tavakolian, Mohammad, Mohammad Sabokrou, and Abdenour Hadid. "AVD: Adversarial Video Distillation." arXiv preprint arXiv:1907.05640 (2019).

**Summary Of The Paper:**

This paper includes two parts. 1) A video frame re-arrangement strategy that transforms a video clip to an super image such that the video can be processed by image model like 2D-CNN; 2) A slightly modified Swin-Transformer that is more suitable to the proposed super image; The proposed method is evaluated on five benchmark datasets to show its effectiveness and efficiency.

**Summary Of The Review:**

Based on the comments in the Main Review part, I tend to reject this paper. The main reasons are:
1) Limittd novelty;
2) Lack of considering the efficiency of the proposed method, i.e. the input image size it too large;
3) Unreasonable model design;

---

> ### Author Response · Authors · 2021-11-23
> **Response to Reviewer utfq**
>
> We thank the reviewer for his/her constructive comments. Below we provide a point by point reply.
>
> **(a) The novelty of the proposed method is limited.**
> While being simple, our idea of using super images for action recognition proposes a new perspective for video understanding without explicit temporal modeling. This is an interesting direction to explore, as pointed out by other reviewers. Our approach also demonstrates very competitive results on multiple datasets.  We expect this work will motivate more ideas to further explore the potential of super images for other video understanding applications such as video object detection and segmentation.
>
>
> **(b) The size of the introduced super image is larger than the video frame.** While the super image has a larger size than a video frame,  our approach has linear computational complexity w.r.t the number of input frames, and it has fewer FLOPs than the majority of approaches compared in Table 3 and 4 under similar settings. On the other hand, the methods [1-3] require a) an encoder-decoder framework to generate a summarized representation; b) an additional image classifier for performing the classification task and c) both RGB and optical flow information to achieve competitive results. Therefore, there is no clear indication that they are more efficient than our approach, if not more costly in computation. Actually, according to Table 5 in [1], the approach of [1] indicates ~10 times more FLOPs than our approach (IFS-3D: 98x30 GFlops v.s SIFAR-B-12: 106x3 GFlops) while still performing worse by 1\% on Kinetics400 (79\% vs 80\%).
>
> **References:**
>
> - [1] Qiu, Zhaofan, et al. “Condensing a Sequence to One Informative Frame for Video Recognition.” Proceedings of the IEEE/CVF International Conference on Computer Vision. 2021.
> - [2] Tavakolian, Mohammad, Hamed R. Tavakoli, and Abdenour Hadid. “Awsd: Adaptive weighted spatiotemporal distillation for video representation.” Proceedings of the IEEE/CVF International Conference on Computer Vision. 2019.
> - [3] Tavakolian, Mohammad, Mohammad Sabokrou, and Abdenour Hadid. “AVD: Adversarial Video Distillation.” arXiv preprint arXiv:1907.05640 (2019).
>
>
> **\(c\) Involvement of the Swin-Transformer seems unreasonable.**
> In our approach, the super image representation converts temporal dependencies into spatial dependencies. Thus, temporal modeling in our case becomes capturing the spatial dependencies among similar patches (or objects). We respectfully disagree with the reviewer that "the local operation can only extract the boundary information of the consecutive frames in the super image". Instead, like other vision transformers, the Swin Transformer is powerful in modeling long-range dependencies between image patches because of its design, i.e. shifted windows and pyramid structure. Note that such dependencies are not necessarily fully captured at a particular or every layer of a network. They can be learnt progressively as the network goes deeper. As shown in Table 2 of the main paper,  our approach using the default window size, i.e SIFAR-B-7, still achieves good performance on both Kinetics400 and SSV2, suggesting that our approach can effectively model temporal dependencies embedded in the super image.
>
> **(d) Missing references.** Thanks for pointing this out. We have added the missing references to our revised manuscript.

---

### Official Review · Reviewer_7Wf5 · 2021-11-02

**Correctness:** 4
**Technical Novelty And Significance:** 3
**Empirical Novelty And Significance:** 3
**Recommendation:** 8
**Confidence:** 3

**Main Review:**

# Strengths

The paper reads well and has almost no typos. The method described is simple and achieves surprising results. The authors have provided extra ablation experiments to evaluate the importance of the grid layout, APE and temporal order of the frames in the grid, as well as activation visualizations using CAM. The experimental setup is clear and results seem convincing.

# Weaknesses

a) It is not entirely clear how the frames are sampled before they are organized in a grid. For example, in other methods such as I3D, consecutive frames are sampled 10 frames apart [A, sections 2.3 and 2.5]. However, the current paper states uniform sampling is used to generate the video input for the models [p.5]. Does it mean the frames are uniformly sampled from the start and ending frames of the entire video, or are they sampled considering a fixed skip with a random starting frame as in I3D?

b) It would seem to me that both the super image representation and the use of a transformer-based architecture are essential in order to achieve such good results due the self-attention mechanism. Have the authors experimented with non-transformer architectures to evaluate the efficacity of the super image representation by itself?

minor) "Our" should be capitalized in second paragraph of page 5, or merged with the previous paragraph.

[A]: Carreira et al, Quo Vadis, Action Recognition? A New Model and the Kinetics Dataset, CVPR 2017.

**Summary Of The Paper:**

The paper deals with action recognition in videos, i.e. detecting to which class a given sequence of frames belongs to. However, the paper proposes to explore whether an image classifier (instead of a video or spatiotemporal-based classifier) would already be enough to accomplish this task. In order to do so, the authors organize the frames from a video into a single image by organizing them into a grid, then proceed to learn them using Swin Transformers (Swin-B) image classification models.

The authors report surprising results which are indeed on-par or higher than the SotA in Kinetics400, MiT, Jester and Diving48 datasets.

**Summary Of The Review:**

The paper presents a simple yet effective idea to transform simpler image classification models into video classification models. Even being simple, the approach manages to achieve surprisingly good results when compared to SotA video-classification models which explicitly handle the temporal dimension. The paper may thus contain findings that should be of interest for the ICLR community.

---

> ### Author Response · Authors · 2021-11-23
> **Response to Reviewer 7Wf5**
>
> We are grateful to the reviewer for the constructive and insightful comments. Below we provide a point by point reply.
>
> **(a) It is not entirely clear how the frames are sampled.** In this work, we adopt uniform sampling [1] to create the input to a SIFAR model. This sampling strategy divides a video into multiple segments of equal length. During training, one frame from each segment is randomly selected to form the input sequence of the model while in test, a frame at a fixed position of each segment (usually the center frame) is picked. It is shown in [2] that uniform sampling is effective on both Kinetics400 and SSV2.
>
> **References:**
> - [1] Temporal Segment Networks: Towards Good Practices for Deep Action Recognition, Limin Wang, Yuanjun Xiong, Zhe Wang, Yu Qiao, Dahua Lin, Xiaoou Tang, Luc Van Gool, ECCV 2016.
> - [2] Deep Analysis of CNN-based Spatio-temporal Representations for Action Recognition, Chun-Fu (Richard) Chen, Rameswar Panda, Kandan Ramakrishnan, Rogerio Feris, John Cohn, Aude Oliva, Quanfu Fan, CVPR 2021.
>
>
> **(b) Experiments with non-transformer architectures** In the paper, we also tried the proposed super image representation with the popular CNN architecture for image classification. The results are listed in Table 7 of the main paper. Our CNN-based SIFAR models outperform traditional CNN-based action models such as I3D and TSM on Kinetics400, but is less effective on SSV2. We further showed in the paper that larger-sized kernels can significantly improve the performance of CNN-based SIFAR models on SSV2.

---

> > ### Comment · Reviewer_7Wf5 · 2021-11-29
> > **Response to response**
> >
> > I thank the authors for their response. It answered my questions and I recommend accepting the paper.

---

### Official Review · Reviewer_JuKq · 2021-11-03

**Correctness:** 4
**Technical Novelty And Significance:** 3
**Empirical Novelty And Significance:** 3
**Recommendation:** 8
**Confidence:** 4

**Main Review:**

In this paper, the authors propose a simple but effective approach for video action recognition by casting the problem as an image recognition task. Different from modeling temporal information, it provides a different perspective to think about the action recognition task.

It provides solid experiments and ablation studies to evaluate the effectiveness of the proposed approach. Transformer-based and CNN-based models are both tested on public benchmarks and good experimental results are reported.



**Summary Of The Paper:**

This paper presents a new approach for video action recognition by casting the problem as an image recognition task. The video clips are rearranged into a super image according to a pre-defined spatial layout.

**Summary Of The Review:**

This paper provides a novel idea for video action recognition. Their claims are well supported by solid experiments and ablation studies. I believe it would inspire others in this research field.

---

> ### Author Response · Authors · 2021-11-23
> **Response to Reviewer JuKq**
>
> We thank the reviewer for the constructive and insightful comments. We are glad that the reviwer likes our idea and believes that "our work would inspire others in this research field".

---

### Author Response · Authors · 2021-11-23
**Summary of Author's Response**

We thank all the reviewers for their inspiring and constructive comments! We are glad that the reviewers found that: **(a)** our proposed method is novel (JuKq), extremely simple (7Wf5, K68r) and easy to implement with one liner in pytorch (K68r); **(b)** our work provides a different perspective to think about the action recognition task (JuKq, K68r) and has potential application field in the future (utfq, K68r); **\(c\)** our experiments are extensive with solid ablation studies (JuKq, utfq, K68r) and interesting visualizations (utfq), showing surprisingly good results when compared to SotA video-classification models (JuKq, 7Wf5); **(d)** our paper contains findings that should be of interest for the ICLR community (7Wf5).

We have addressed all the concerns that the reviewers posed with additional experimental comparisons and clarifications. All of these additional experiments and suggestions have been added to the updated PDF (changes are highlighted in red).

---

### Decision · Program_Chairs · 2022-01-20

**Decision:**

Accept (Poster)

**Comment:**

This paper regards video understanding as an image classification task, and reports promising performance against state of the arts on several standard benchmarks. Though the method is quite simple, it achieves good results. The visualization in this paper also provides good insight. All reviewers give positive recommendations for this paper.